# Development of a CYP2D6-enhanced HepaRG cell model with improved CYP2D6 metabolic capacity

Chizuka Obara[1], Yohei Iizaka[2], Akari Mine[1], Yojiro Anzai[2], Masako Tada[1†‡], Shinpei Yamaguchi[1‡*]

1 Stem Cells and Reprogramming Laboratory, Department of Biology, Faculty of Science, Toho University, Funabashi, Chiba, Japan, 2 Department of Microbiology, Faculty of Pharmaceutical Sciences, Toho University, Japan

† Deceased 21 October 2023.
‡ MT and SY shared senior co-authorship.
* shinpei.yamaguchi@sci.toho-u.ac.jp

## Abstract

Drug-induced liver injury is a major concern in drug development, and the limitations of animal models have driven interest in human-derived in vitro models for toxicity assessment. HepaRG cell-based liver models are promising due to their ability to differentiate into hepatocyte-like cells that highly express Cytochrome P450 (CYP) 3A4; however, their low CYP2D6 expression limits their utility in representing human liver metabolism. CYP2D6 is a polymorphic enzyme responsible for inter-individual variability in drug metabolism and is associated with adverse drug reactions. To address this limitation, the present study aimed to enhance CYP2D6 expression in undifferentiated HepaRG cells, successfully generating multiple transgenic cell lines with varying levels of CYP2D6 activity. We developed FLAG-tagged CYP2D6 and IRES-green fluorescent protein (GFP)-co-expressing CYP2D6 HepaRG cells, enabling CYP2D6 expression to be monitored via FLAG and GFP signals. These cells exhibited a remarkable 5- to 8,000-fold increase in CYP2D6-mediated bufuralol metabolism. Notably, the CYP2D6 expression level in a CYP2D6-iGFP cell line was comparable to that observed in human liver tissue. Furthermore, enforced CYP2D6 expression showed a tendency to reduce the cytotoxicity of perhexiline, a drug primarily metabolized by CYP2D6. Among the five transgenic cell lines we developed, two cell lines retained the differentiation potential of HepaRG cells, giving rise to CYP3A4-positive hepatocyte-like cells. Moreover, we demonstrated that CYP2D6-iGFP HepaRG cells retain transgene expression following differentiation into hepatocyte-like cells in vitro. While some transgenic HepaRG cell lines exhibited limited differentiation potential, the CYP2D6-enhanced cells will serve as an effective cell model for in vitro studies of drug metabolism and toxicity, especially for compounds that are metabolized by CYP2D6.

**Data availability statement:** The main data supporting the findings of this study are included within the manuscript and its Supporting Information files. Additional raw data, including numerical values used for statistical analyses and microscopy source files, are available from the Zenodo repository (DOI: 10.5281/zenodo.17411014 ).

**Funding:** Toho University Grant for Research Initiative Program. The funders had no role in study design, data collection and analysis, decision to publish, or preparation of the manuscript.

**Competing interests:** The authors have declared that no competing interests exist.

# 1 Introduction

Drug-induced liver injury is the leading cause of post-marketing withdrawal of approved drugs [1]. Traditionally, drug safety assessments have relied on animal studies and human clinical trials. However, animal-based toxicity tests correlate poorly with human clinical outcomes, and approximately half of the medicines in the "WITHDRAWN" database have been withdrawn or discontinued post-approval due to safety concerns [2]. Due to fundamental differences in physiological functions and drug metabolism patterns between humans and animals, animal-based toxicity tests frequently fail to detect human-specific toxicity [3,4]. Consequently, there is an urgent need for a simple *in vitro* human liver model capable of effectively predicting drug-induced hepatotoxicity during early-stage drug development.

Since the majority of drugs are metabolized by cytochrome P450 (CYP) enzymes within the liver, primary cultured human hepatocytes have been extensively utilized as an in vitro human liver model for toxicity and metabolism studies. However, primary cultured human hepatocytes rapidly lose CYP activity, and batch-to-batch variations result in poor reproducibility [5,6].

HepaRG cells have emerged as a widely studied model system for human hepatocytes due to their unique ability to differentiate into both hepatocyte-like and cholangiocyte-like cells following treatment with hydrocortisone and dimethyl sulfoxide (DMSO) [7]. Although originally derived from a hepatocellular carcinoma in a patient with chronic hepatitis C, HepaRG cells preserve their differentiation potential while maintaining expression profiles of key hepatic enzymes, including some cytochrome P450 (CYP) isoforms, phase II metabolizing enzymes, drug transporters, and nuclear receptors. Moreover, these cells successfully recapitulate essential hepatocyte functions, such as albumin synthesis and glycogen storage, making them an invaluable tool for hepatotoxicity studies and drug metabolism research [7–10]. Notably, differentiated HepaRG cells express exceptionally high levels of CYP3A4, the predominant drug-metabolizing enzyme responsible for processing over 50% of human pharmaceuticals. In our previous work, we established a dual-color reporter HepaRG cell line that enables real-time visualization of CYP3A4 expression dynamics, and revealed their remarkable phenotypic plasticity [11,12]. This system allows for a CYP3A4 induction assay in response to compounds such as rifampicin, providing a means to evaluate the enzyme-inducing potential of candidate drugs. These capabilities are notably absent in other liver tumor-derived cell lines, such as HepG2, positioning HepaRG cells as an ideal model for investigating drug-induced hepatotoxicity and hepatic metabolism.

However, the expression level of another critical CYP enzyme, CYP2D6, is significantly lower in HepaRG cells than in human liver biopsy samples and primary human hepatocytes [13,14]. Additionally, the CYP2D6 activity of HepaRG cells is markedly lower than that of suspended primary human hepatocytes, which are used to predict in vivo drug clearance [15].

CYP2D6 metabolizes approximately 25% of commonly prescribed drugs across various medical fields, including psychiatry, pain management, oncology, and cardiology [16,17]. Its expression and activity vary widely among individuals due to genetic

polymorphisms, including mutations, copy number variations, and alternative splicing [18]. Based on these genetic variations, humans can be categorized into four distinct CYP2D6 metabolizer phenotypes: poor metabolizer (PM), intermediate metabolizer (IM), extensive metabolizer (EM), and ultrarapid metabolizer (UM) [19]. These phenotypic differences have significant clinical implications. For example, patients with PM or IM phenotypes face an increased risk of severe hepatotoxicity and neuropathy when treated with perhexiline, an antianginal medication [19]. Conversely, UMs demonstrate accelerated conversion of codeine to morphine, potentially resulting in morphine intoxication [20]. Consequently, individuals with enhanced or diminished CYP2D6 enzyme activity may experience compromised therapeutic efficacy, elevated hepatotoxicity risk, or increased susceptibility to adverse drug reactions when prescribed medications that are substrates of CYP2D6.

HepaRG cells retain proliferative capacity in their undifferentiated state, offering a practical advantage over primary human hepatocytes by allowing easier expansion to the quantities required for screening assays. However, their endogenous CYP2D6 activity is relatively low and does not accurately represent the typical enzymatic profile of human liver tissue. Moreover, the functional consequences of elevated CYP2D6 expression in HepaRG cells remain largely unexplored. To establish a more physiologically relevant platform for screening compounds metabolized by CYP2D6, we engineered undifferentiated HepaRG cells with enhanced CYP2D6 function.

## 2 Materials and methods

### 2.1 Cell culture

Undifferentiated HepaRG cells were obtained from Biopredic International (St. Grégoire, France) and cultured according to the manufacturer's protocol. Wild-type (WT) HepaRG cells with passage numbers within P19 were used for all experiments. For differentiation into hepatocyte-like cells, proliferative HepaRG cells were seeded in 24-well plates at different cell densities ($2.5 \times 10^4$, $5 \times 10^4$, and $10 \times 10^4$ cells/cm$^2$), respectively. After 14 days, the cells were cultured in 1% DMSO medium for 2 days, and then further cultured in medium supplemented with 1.7% DMSO supplementation for 12 days [12,21,22].

### 2.2 Enhancement of CYP2D6 activity in HepaRG cells

The FLAG-CYP2D6 plasmid with the human cytomegalovirus (CMV) promoter was constructed by subcloning the *CYP2D6* open reading frame (ORF) into the p3×FLAG epitope tag (FLAG)-CMV10 plasmid (pCMV-FLAG-*CYP2D6*) (Sigma-Aldrich). The pCMV-*CYP2D6*-IRES-green fluorescent protein (GFP) expression vector was previously constructed in our laboratory [23]. CYP2D6 expression vectors were transfected into undifferentiated HepaRG cells according to the previous study with minor modifications. Briefly, linearized FLAG-*CYP2D6* or *CYP2D6*-iGFP/ phosphoglycerate kinase promoter–neomycin resistance (PGK-neo) vectors were transfected into HepaRG cells using Lipofectamine-LTX (Thermo Fisher Scientific), respectively. After 2 days of culture, stably transfected cells were selected in 0.5 mg/mL G418-containing medium (Fujifilm Wako Pure Chemical Corp., Osaka, Japan). The G418-resistant colonies were pooled and subjected to further selection in a medium 0.2 mg/mL G418-containing medium. Subsequently, the cells were cultured in growth medium supplemented with G418 in order to maintain stable CYP2D6 expression.

### 2.3 Real-time quantitative RT-PCR analysis

Total RNA was isolated from HepaRG cells using the RNeasy Mini Kit (Qiagen, Venlo, Netherlands). cDNA synthesis was performed with 0.5 µg of total RNA using random hexamer primers and SuperScript III reverse transcriptase (Thermo Fisher Scientific). Human adult liver cDNA was purchased from TAKARA Bio Inc. Gene amplification was measured using the CFX384 Touch Real-Time PCR detection system (Bio-Rad) and quantified using the standard curve-based method. The primer sets used for the assays are listed in Table 1.

**Table 1. Primer sets used for RT-qPCR.**

| Sets | Gene | Size | Primer name | Sequence (5' to 3') |
|---|---|---|---|---|
| 1 | *CYP2D6* | 95 bp | hqCYP2D6-F | GAAGGAGGAGTCGGGCTTT |
| | | | hqCYP2D6-R | TTTGGAAGCGTAGGACCTTG |
| 2 | *CYP1A2* | 167 bp | hqCYP1A2-F-2 | TAGCGATGAGATGCTCAGCC |
| | | | hqCYP1A2-R-2 | CTGTTTTCTGCAGGAACCACAG |
| 3 | *ACTB* | 76 bp | ACTB-F | ATTGGCAATGAGCGGTTC |
| | | | ACTB-R | GGATGCCACAGGACTCCAT |

## 2.4 Western blotting analysis

HepaRG cells were lysed with RIPA buffer (Tokyo Chemical Industry Co., Ltd., Tokyo, Japan) containing a protease inhibitor cocktail (cOmplete Mini, Roche Diagnostics, Basel, Switzerland). After 30 min of incubation on ice, the cell lysate was centrifuged for 20 min at 10,000×g, and the supernatant was collected as the soluble protein fraction. The protein concentration of the samples was quantified using a Bicinconic Acid (BCA) protein assay (Thermo Fisher Scientific). Protein samples were separated by sodium dodecyl sulfate–polyacrylamide gel electrophoresis (SDS-PAGE), transferred to polyvinylidene difluoride (PVDF) membranes, and incubated with Blocking One (Nacalai Tesque) for 30 min. The membranes were then incubated with primary antibodies: anti-CYP2D6 (Abcam, ab185625, 1:3000) and anti-β-actin (Abcam, ab8226, 1:1000). After washing three times with Tris-buffered saline with 0.1% Tween-20 (TBS-T), the membranes were incubated with peroxidase conjugated secondary antibodies (sheep anti-mouse IgG or donkey anti-rabbit IgG, GE Healthcare) for 60 min. Signals were visualized using Clarity Western ECL Substrate (Bio-Rad) and ChemiDoc MP Imaging System (Bio-Rad). The raw images of Western blotting are shown in S1 Fig. All data were analyzed using ImageJ software, and CYP2D6 band intensities were normalized to those of b-actin (S1 Fig).

## 2.5 Immunofluorescence staining

Cells were cultured on glass-bottom 35 mm dishes (Matsunami Glass, Osaka, Japan) or Sumilon cell-discs LF1 (Sumitomo Bakelite Co. Ltd, Tokyo, Japan), fixed with 4% paraformaldehyde/phosphate-buffered saline (PBS) for 10 min, and washed with PBS. The fixed cells were then permeabilized with 0.1% Triton X-100 for 10 min and washed three times with PBS. After blocking with 1% bovine serum albumin (BSA)/PBS Blocking buffer for 30 min, the cells were incubated overnight at 4°C with primary antibodies: anti-CYP2D6 (Abcam, ab185625, 1:500), anti-FLAG (Sigma-Aldrich, F1804, 1:200), anti-CYP3A4 (Enzo Life Sciences, BML-CR3340–0100, 1:300) or anti-human Albumin (Takara Bio Inc, M226, 1:500). Following three PBS washes, the cells were incubated for 1 h with 4′,6-diamidino-2-phenylindole (DAPI) (Sigma-Aldrich) and secondary antibodies: Alexa Fluor 568-conjugated goat anti-rabbit IgG or Alexa Fluor 488-conjugated goat anti-mouse IgG (Thermo Fisher Scientific, 1:500). After three times PBS washes, cells were observed using a CFI Plan Fluor DL 10× objective on a BZ-X710 (Keyence, Osaka, Japan). A negative control experiment was performed using only the secondary antibody without a primary antibody.

## 2.6 Periodic Acid-Schiff (PAS) staining

PAS staining of differentiated HepaRG cells was performed using a PAS staining kit (Muto Chemical, Tokyo, Japan) according to the manufacturer's instructions.

## 2.7 CYP2D6 enzyme activity analysis

Cells were seeded onto 6-well plates at $2.5 \times 10^4$ cells/cm$^2$ and cultured for 5 days. After washing with PBS, the cells were incubated with 10 µM bufuralol (Cayman Chemical, Michigan, USA), a substrate for CYP2D6, in phenol red-free MEM

for 1 h at 37°C. The supernatant was collected and diluted threefold in ice-cold methanol. Cellular debris was removed by centrifugation at 20,600 × g for 10 min, and the supernatant was stored at −20°C for further analysis. Five µL of the supernatant was injected into a Liquid Chromatograph-tandem Mass Spectrometer (LC-MS/MS) system using an LCMS-8040 (Shimadzu Corporation, Kyoto, Japan) fitted with an STR ODS-II column (2.0 mm i.d. × 150 mm; Shinwa Chemical Industries, Tokyo, Japan) maintained at 35°C. Mobile phase consisted of 0.1% formic acid in water (A) and acetonitrile (B). The flow rate was set to 0.2 mL/min, with a gradient from 20% B to 95% B over 2 min, held at 95% B from 2.1 to 5 min, and returned to 20% B from 5.1 to 7 min, followed by re-equilibration at 20% B until 15 min. The mass spectrometer was operated in multiple-reaction monitoring (MRM) mode using positive ion electrospray under the following operating conditions: heat-block temperature, 500°C; desolvation line temperature, 300°C; nebulizer gas flow rate, 3.0 L/min; drying gas flow rate, 15 L/min; collision-induced dissociation gas pressure, 230 kPa, and ion spray voltage, 4.5 kV. The multiple-reaction monitoring (MRM) transitions (*m/z* of precursor ion/*m/z* of product ion) for 1'-hydroxy (1'-OH) bufuralol was 278.0 (Q1)/186.0 (Q3). 1'-OH bufuralol in the reaction mixture was quantified using linear calibration curves generated from standard solutions, and concentrations were normalized to the protein content per well. In this assay, we have been confirmed that increasing bufuralol concentrations led to greater 1'-OH bufuralol production, indicating that CYP2D6 metabolism is substrate concentration-dependent (S2 Fig).

## 2.8 Cytotoxicity assay

WT HepaRG cells ($2.5 \times 10^4$ cells/cm$^2$) or transgenic HepaRG cells ($5 \times 10^4$ cells/cm$^2$) were seeded into 96-well plates and cultured in growth medium. After a 4-day pre-culture, the cells were incubated with the compounds in G418-free medium for 24 hours. Based on the Drug-Induced Liver Injury (DILI) Rank dataset of 1,036 Food and Drug Administration (FDA)-approved drugs, we selected five compounds for investigation: Perhexiline (Sigma-Aldrich), Amiodarone HCl (Selleck Chemicals, Houston, TXA, USA), Troglitazone (Selleck Chemicals), Tramadol HCl (Sigma-Aldrich), and Trazodone HCl (Selleck Chemicals) [24]. The details of the compounds used in the assays are provided in Table 2. Cellular viability was assessed using the CellTiter-Glo Luminescent Cell Viability Assay (Promega Corporation, Madison, WI). Luminescence was measured with a Varioskan LUX Multimode Microplate Reader (Thermo Fisher Scientific).

## 2.9 Statistical analysis

Data are presented as mean with individual measurements, or mean ± standard deviation (SD). The assumptions of normality and homogeneity of variance were assessed using the Kolmogorov-Smirnov test and Bartlett's test, respectively. For multiple comparisons versus the common WT control, we used a one-factor linear model with pre-specified Dunnett contrasts. Two-sided tests with α = 0.05 were used, and Dunnett-adjusted p values are reported (significant if adjusted p < 0.05). Statistical analysis was performed using EZR software (Saitama Medical Centre, Jichi Medical University, Saitama, Japan) [25].

**Table 2. List of the tested compounds for the cell toxicity assay.**

| Compounds | v-DILI concern[a] | Label[b] | Major metabolic CYP enzymes | Concentrations (µM) |
|---|---|---|---|---|
| Perhexiline | Most-DILI-concern | WD | CYP2D6[1] | 5-22 |
| Troglitazone | Most-DILI-concern | WD | CYP3A4[2] | 67.5-200 |
| Amiodarone | Most-DILI-concern | BW | CYP3A4[3] | 12.5-60 |
| Trazodone | Less-DILI-Concern | AR | CYP3A4[4] | 0.6-28 |
| Tramadol | Ambiguous DILI-concern | AR | CYP2D6, CYP3A4[5] | 3.1-100 |

[a]verified-DILI-concern from the Drug Induced Liver Injury (DILI) rank dataset of 1,036 Food and Drug Administration (FDA)-approved drugs

[b]Drug labels in the DILIrank dataset: WD withdrawn, BW black box warning, AR adverse reactions

[1]Ashrafian et al. (2007), [2]Kassahun et al. (2001), [3]Zahno et al. (2011), [4]Rotzinger et al. (1998), [5]Miotto et al., (2017)

## 3 Results

### 3.1 Establishment of FLAG-CYP2D6 and CYP2D6-iGFP undifferentiated HepaRG cells

To enhance CYP2D6 activity in HepaRG cells, we constructed two vectors: a FLAG-CYP2D6-expressing vector and a *CYP2D6*-IRES-GFP (CYP2D6-iGFP)-expressing vector, both under transcriptional control of the CMV promoter (Fig 1A). The internal ribosome entry site (IRES) element enables the translation of both proteins from a single mRNA. Initial attempts to generate clonal (single-cell-derived) FLAG-CYP2D6-expressing cell lines using G418-based selection were unsuccessful, as the resulting clones failed to maintain proliferation after several passages. Therefore, we established polyclonal lines and obtained three FLAG-CYP2D6 cell lines (#1, #2, and #3). The stable expression of FLAG-tagged CYP2D6 did not alter the morphology of HepaRG cells (Fig 1B). Additionally, we successfully established two polyclonal CYP2D6-iGFP HepaRG cell lines (Fig 1C). Both lines maintained stable GFP expression throughout successive passages; however, #1 CYP2D6-iGFP HepaRG cells exhibited higher GFP fluorescence intensity than #2 CYP2D6-iGFP HepaRG cells.

### 3.2 Subcellular localization and protein expression

Immunofluorescence staining against FLAG revealed that FLAG-CYP2D6 localized to the endoplasmic reticulum (ER)-rich perinuclear regions (Fig 1D), consistent with the reported subcellular distribution of native or transgenic CYP2D6 (Yamamoto et al., 1993; Islam et al., 2024). In CYP2D6-iGFP HepaRG cells, CYP2D6 similarly localized to ER-rich perinuclear regions in GFP-positive cells (Fig 1D). We further investigated the expression levels of CYP2D6 protein in each HepaRG cell line using Western blotting analysis (Fig 1E). FLAG-tagged CYP2D6 was detected only in FLAG-CYP2D6 HepaRG cells, while endogenous CYP2D6 was detected in both FLAG-CYP2D6 and WT HepaRG cells. Among the FLAG-CYP2D6 clones, the highest FLAG-tagged CYP2D6 protein expression was observed in clone #3, followed by clone #1, with the lowest expression observed in clone #2.

### 3.3 Enzymatic activity assessment

To confirm the catalytic activity of FLAG-CYP2D6 HepaRG cells, we measured CYP2D6 enzymatic activity. HepaRG cells were incubated with bufuralol, a substrate primarily metabolized by CYP2D6, and 1′-hydroxybufuralol (1'-OH bufuralol), a CYP2D6-mediated hydroxylation product, was quantified using LC-MS/MS analysis (Fig 1F). The average 1'-OH bufuralol levels were as follows: $0.013 \pm 0.003$ nmol min$^{-1}$ mg$^{-1}$ protein in WT, $1.76 \pm 0.11$ in #1 FLAG-CYP2D6, $0.23 \pm 0.07$ in #2 FLAG-CYP2D6, and $5.11 \pm 1.9$ in #3 FLAG-CYP2D6 HepaRG cells. Consistent with previous reports, the catalytic activity of CYP2D6 was extremely low in WT HepaRG cells, despite detectable protein levels in Western blot analysis [15,26,27]. These results indicate that the CYP2D6 catalytic activity of #1, #2, and #3 FLAG-CYP2D6 HepaRG cells was primarily mediated by exogenous CYP2D6, exhibiting approximately 135-, 18-, and 393-fold higher activity, respectively, compared to parental WT HepaRG cells. The increase in 1'-OH bufuralol formation in cells expressing high levels of FLAG-tagged CYP2D6 confirmed a positive correlation between enzyme activity and transgene expression.

In CYP2D6-iGFP cell lines, Western blotting analysis demonstrated that CYP2D6 protein levels correlated with GFP intensity (Fig 1C and 1G). CYP2D6 catalytic activity was evaluated by quantifying 1'-OH bufuralol formation using LC-MS/MS (Fig 1H). The average 1'-OH bufuralol production ranged from $0.013 \pm 0.003$ nmol min$^{-1}$ mg$^{-1}$ protein in WT HepaRG cells to $112 \pm 29$ in #1 CYP2D6-iGFP HepaRG cells, and $0.073 \pm 0.009$ in #2 CYP2D6-iGFP HepaRG cells, representing approximately 8,615-fold and 5.6-fold increases, respectively.

Finally, we compared the gene expression level of CYP2D6 in the newly established cell lines with that in adult human liver tissue. Among the five cell lines generated in this study, #1 CYP2D6-iGFP , which exhibited the highest CYP2D6 activity, showed a CYP2D6 expression level comparable to that in human liver tissue (Fig 1I).

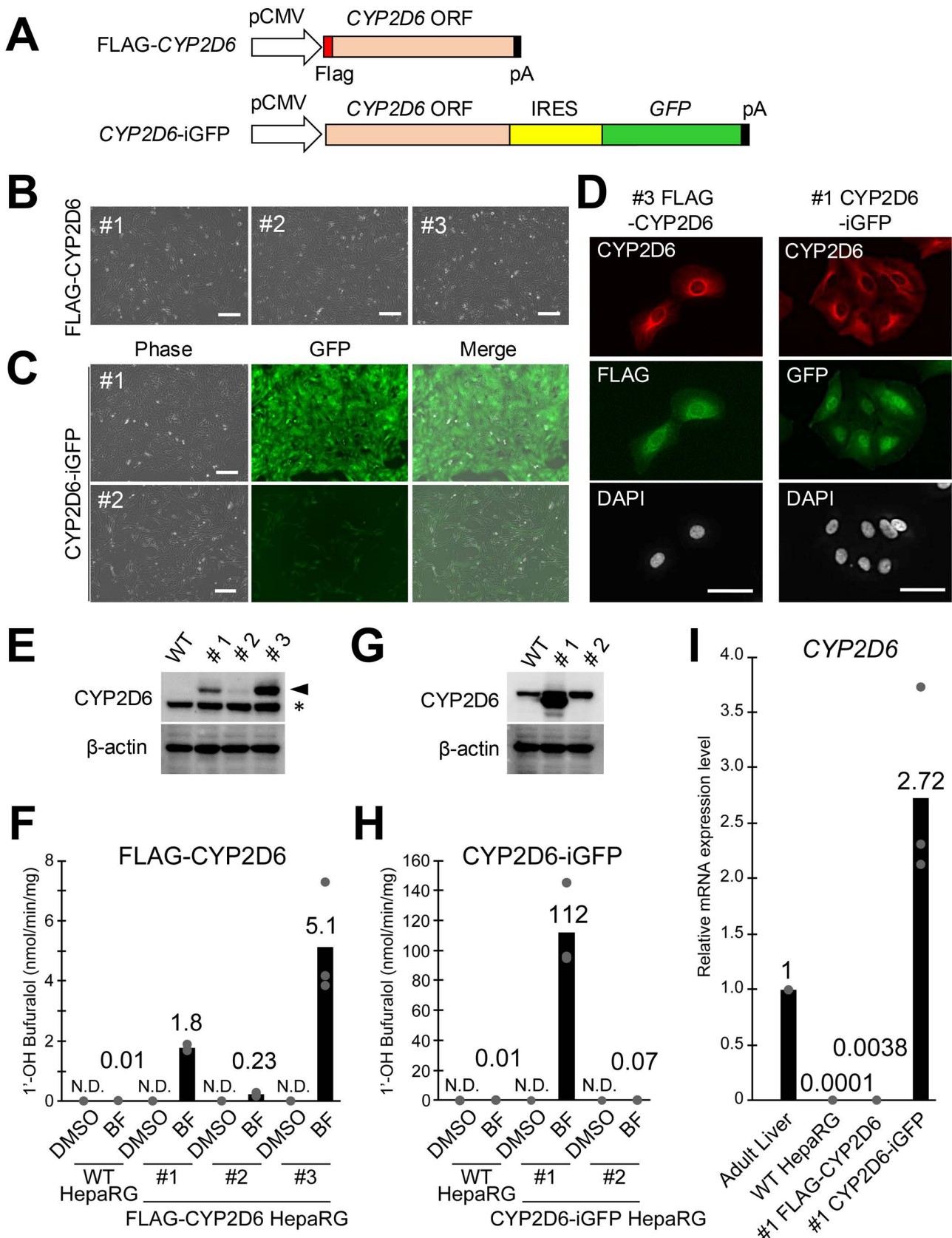

**Fig 1. Establishment and characterization of FLAG-CYP2D6 and CYP2D6-iGFP undifferentiated HepaRG cells. (A)** Transgene structures for the expression of the exogenous FLAG-tagged CYP2D6 protein or the IRES-based co-expression of the human hepatic CYP2D6 and GFP proteins. **(B)** Phase contrast images of the FLAG-CYP2D6 HepaRG cell line. Scale bar, 200 μm. **(C)** Representative phase contrast and fluorescence images of CYP2D6-iGFP HepaRG cells. Scale bar, 200 μm. **(D)** Representative immunofluorescence images showing the subcellular localization of CYP2D6 in #3 FLAG-CYP2D6 and #1 CYP2D6-iGFP HepaRG cells. Scale bar, 50 μm. **(E)** Western blot analysis of CYP2D6 expression in FLAG-CYP2D6 HepaRG cells. FLAG-tagged CYP2D6 protein was indicated by the arrowhead. An asterisk indicates the bands corresponding to endogenous CYP2D6 protein. Uncropped Western blot images are shown in S1 Fig. **(F)** CYP2D6 enzyme activity in FLAG-CYP2D6 HepaRG cells. The activity of CYP2D6-catalyzed bufuralol hydroxylation was measured by the detection of 1'OH bufuralol using LC-MS/MS analysis. "BF" indicates the bufuralol-treated group. DMSO was used as a vehicle control without a substrate. Each dot represents an individual replicate (n = 3). **(G)** Western blot analysis of CYP2D6 expression in WT #1 and #2 CYP2D6-iGFP HepaRG cells. Uncropped Western blot images are shown in S1 Fig. **(H)** CYP2D6 enzyme activity in CYP2D6-iGFP HepaRG cell lines. The activity of CYP2D6-catalyzed bufuralol hydroxylation activity was measured by the detection of 1'OH bufuralol using LC-MS/MS analysis. "BF" indicates the bufuralol-treated group. DMSO was used as a vehicle control. Each dot represents an individual replicate (n = 3). **(I)** RT-qPCR analysis of CYP2D6 mRNA levels in adult liver tissue, #1 FLAG-CYP2D6 and #1 CYP2D6-iGFP HepaRG cells. Each dot represents an individual replicate (n = 3).

## 3.4 Enhanced CYP2D6 expression in undifferentiated HepaRG cells attenuates perhexiline-induced cytotoxicity

To investigate the physiological significance of enhanced CYP2D6 expression, we assessed the cytotoxic effects of five drugs from the Drug-Induced Liver Injury (DILI) Rank dataset—perhexiline, troglitazone, amiodarone, trazodone, and tramadol—all known for their potential to cause drug-induced liver injury, in both WT and CYP2D6-enhanced cells (Table 2).

The antianginal drug perhexiline, a hepatotoxic compound not subject to metabolic activation, is primarily metabolized by CYP2D6, which catalyzes the hydroxylation of perhexiline to the less-toxic *cis*-OH-perhexiline (Fig 2A). The viability of CYP2D6-overexpressing and WT HepaRG cells was assessed after 24 hours of culture at various concentrations of perhexiline (5–22 μM). At concentrations of 13 to 16.9 μM, WT HepaRG cells showed significant cytotoxicity in response to perhexiline, whereas #1 CYP2D6-iGFP HepaRG cells displayed only mild cytotoxicity (Fig 2B). Notably, the GFP signal was lost due to cell death induced by 22 μM perhexiline in CYP2D6-iGFP HepaRG cells (Fig 2B). A cellular ATP assay revealed a concentration-dependent decrease in viability for WT HepaRG cells at concentrations ranging from 10 to 16.9 μM, with no similar effects observed in #1 CYP2D6-iGFP HepaRG cells (Fig 2C). The estimated $IC_{50}$ for perhexiline in WT HepaRG cells was 14.5 μM, compared to 18 μM in #1 CYP2D6-iGFP HepaRG cells, indicating attenuated perhexiline toxicity (Fig 2D). Similar results were observed in #1 FLAG-CYP2D6 HepaRG cells (S3 Fig). Compared to WT controls, cell viability tended to be higher in #3 FLAG-CYP2D6 cells, although no statistically significant differences were observed (S3 Fig). These findings suggest that CYP2D6 overexpression in undifferentiated HepaRG cells enhances cellular protection against perhexiline-induced cytotoxicity.

## 3.5 Specificity of CYP2D6-Mediated protection in undifferentiated HepaRG cells

To investigate the specificity of the protective effect of CYP2D6 in undifferentiated transgenic cells, we examined their sensitivity to CYP3A4 substrates. As expected, troglitazone and amiodarone induced dose-dependent cytotoxicity in both WT and CYP2D6-iGFP HepaRG cells, consistent with the fact that they are not primarily metabolized by CYP2D6. In contrast, trazodone and tramadol had no significant effect on cell viability under the current dosing protocol (S4 Fig). As it is known that drugs other than perhexiline are mainly metabolized by CYP3A4, there was no difference in drug sensitivity between WT cells and CYP2D6-enhanced cells (Table 2).

## 3.6 Differentiation capacity of CYP2D6-enhanced HepaRG cells

To evaluate whether CYP2D6 expression-enhanced HepaRG cells retained their differentiation potential into hepatocyte-like cells, FLAG-CYP2D6 and CYP2D6-iGFP HepaRG cells were cultured in a DMSO-containing induction medium for 14 days (Fig 3A). One hallmark of differentiated hepatocytes is the synthesis and storage of glycogen. To assess differentiation efficiency, PAS staining, which detects cytoplasmic glycogen granules, was performed on differentiated cells.

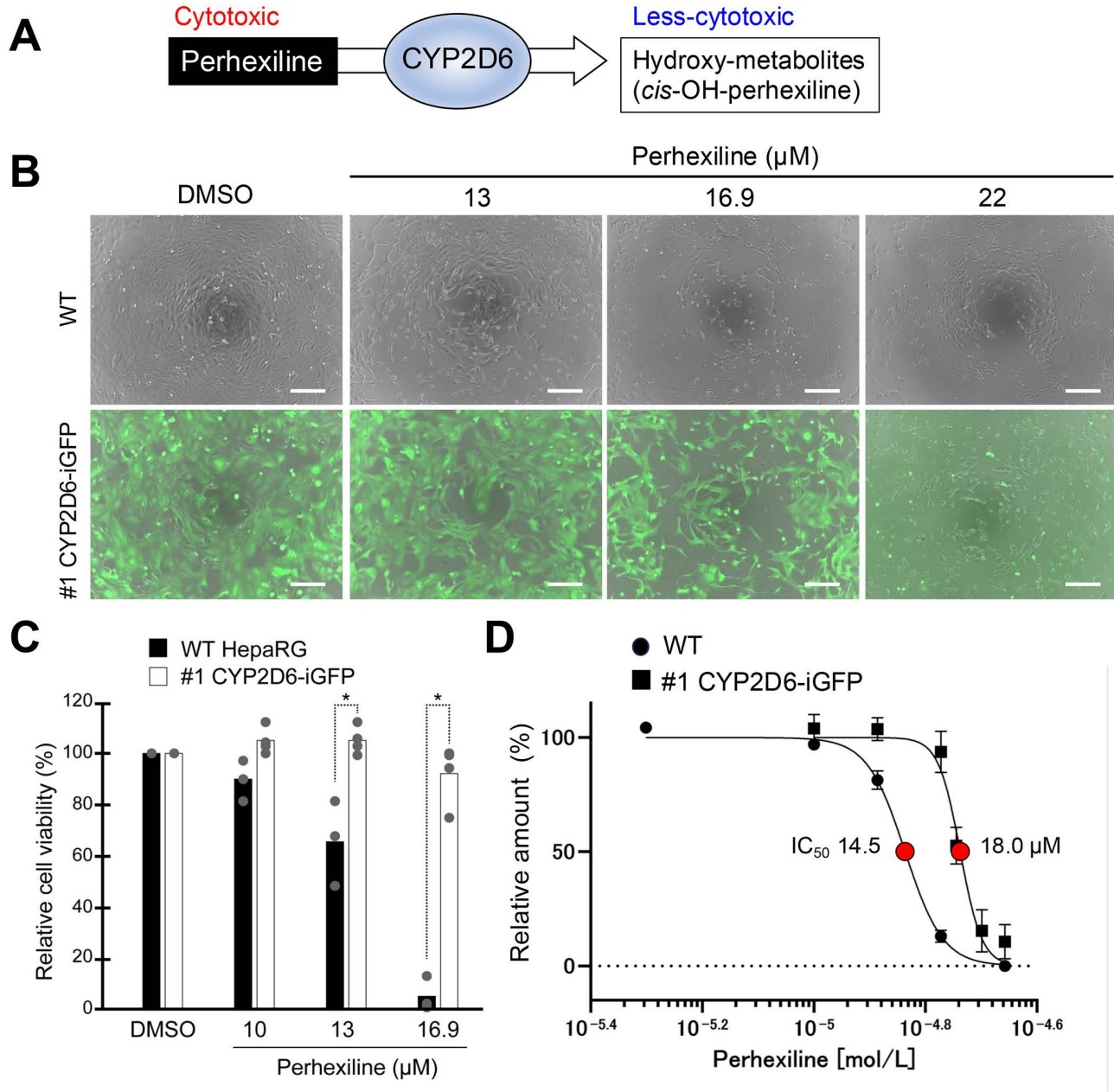

**Fig 2. A cytotoxicity assay of perhexiline in CYP2D6-iGFP undifferentiated HepaRG cells. (A)** Schematic representation of perhexiline detoxification by CYP2D6 activity. **(B)** Representative images of WT and #1 CYP2D6-iGFP HepaRG cells after 24-hour exposure to perhexiline. Merged images of GFP and phase contrast are shown. Scale bar, 200 μm. **(C)** Relative cell viability of perhexiline treated cells assessed by an ATP-based cell viability assay. The results in perhexiline-treated cells were normalized to those in DMSO controls. Each dot represents an individual replicate (n = 3-4). P value calculated from Dunnett's multiple comparison test. *$P < 0.05$. **(D)** Dose-response curves of relative cell viability in perhexiline-treated cells. Each dot represents an individual value. The estimated $IC_{50}$ value is indicated by a red closed circle. n = 3.

Although PAS-positive cells were less frequent than in WT cells, they were detected in both #1 FLAG-CYP2D6 and #1 CYP2D6-iGFP HepaRG cells (Fig 3B and S5 Fig). Increasing the number of cells undergoing differentiation improved the differentiation efficiency in both #1 FLAG-CYP2D6 and #1 CYP2D6-iGFP HepaRG cells. The PAS-positive signal in WT,

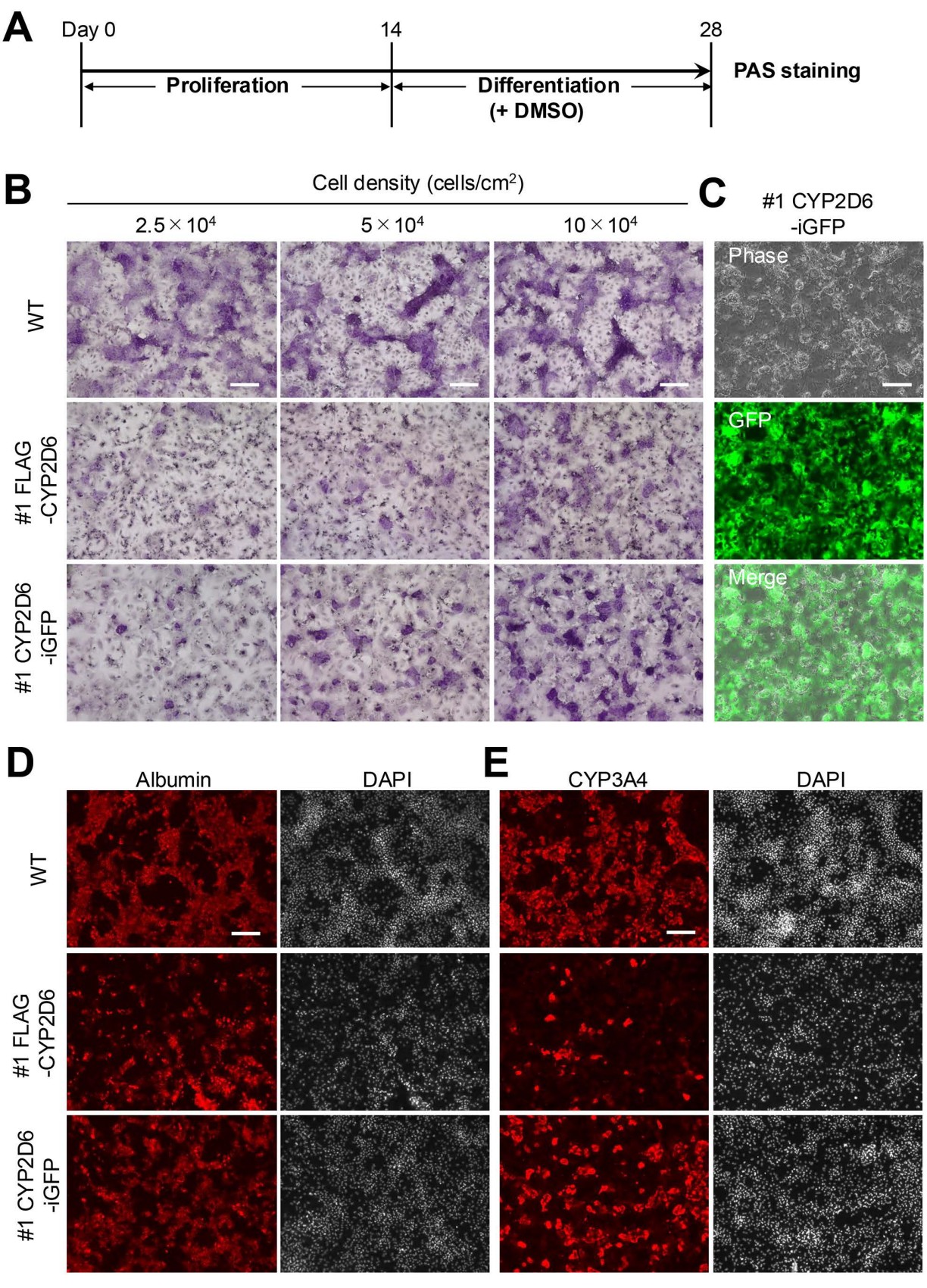

**Fig 3. Differentiation assay of FLAG-CYP2D6 and CYP2D6-iGFP HepaRG cells. (A)** Schematic representation of the differentiation induction process for HepaRG cells. Periodic acid-Schiff (PAS) staining was performed at the end of 28-day differentiation. **(B)** Representative images of PAS staining of differentiated HepaRG cells. Scale bar, 200 μm. **(C)** Representative fluorescence microscopy images showing GFP expression in differentiated #1 CYP2D6-iGFP HepaRG cells. Scale bar, 200 μm. **(D, E)** Immunofluorescence analysis of hepatic marker expression in differentiated cells. Representative images showing Albumin (D) and CYP3A4 (E) expression. Nuclei were counterstained with DAPI. Scale bars, 200 μm.

#1 FLAG-CYP2D6, and #1 CYP2D6-iGFP HepaRG cells confirmed the presence of glycogen (S5 Fig). In contrast, cells derived from #2 and #3 FLAG-CYP2D6, as well as #2 CYP2D6-iGFP HepaRG cells, did not exhibit a clear PAS-positive signal (S5 Fig).

### 3.7 Maintenance of transgene expression after differentiation

After induction, GFP fluorescence intensity in #1 CYP2D6-iGFP HepaRG cells remained stable, indicating that CYP2D6 expression was maintained during the differentiation process (Fig 3C). Next, we investigated the expression of hepatocellular markers, albumin and CYP3A4, in the differentiated cells and detected immunepositive signals in the hepatocyte-like cells (Fig 3D and 3E). CYP2D6 expression in differentiated transgenic cell lines remained significantly higher than that in WT cells, while CYP1A2 expression showed a decreasing trend (S6 Fig). Notably, in #1 CYP2D6-iGFP HepaRG cells, differentiated cells co-expressing GFP and CYP3A4 were observed (S6 Fig). These results demonstrated that the #1 CYP2D6-iGFP HepaRG cell line is capable of differentiating into hepatocyte-like cells while maintaining transgene expression.

## 4 Discussion

Recently, the FDA Modernization Act 2.0 in the U.S. removed the mandatory requirement for animal testing in drug approval processes. This change allows alternative methods—such as cell-based assays using human cells, organoids, organ-on-a-chip systems, and advanced artificial intelligence techniques—if they demonstrate equivalence to animal testing [28]. Beyond concerns about animal welfare, this shift reflects a growing recognition of the limitations of animal models in predicting human responses. Human cell-based systems, which more closely mimic human physiology, are expected to enhance the sensitivity and accuracy of drug toxicity detection. In particular, human hepatic cells are valuable for assessing drug metabolism and liver toxicity. In Japan, the Pharmaceuticals and Medical Devices Agency (PMDA) also promotes the appropriate use of non-animal, human cell-based approaches when scientifically justified, and our CYP2D6-enhanced HepaRG system aligns with this direction [29]. Likewise, the European Medicines Agency recognizes the value of human *in vitro* models and encourages their use alongside other evidence to improve human relevance.

In this context, HepaRG cells have emerged as a promising tool for *in vitro* liver models. Their ability to express key drug-metabolizing enzymes, including CYP family isoforms, makes them a valuable alternative to primary human hepatocytes. However, current HepaRG cell lines derived from a single donors are unable to capture the polymorphic variations observed in human CYP2D6 activity. The *CYP2D6* gene in HepaRG cells contains at least one CYP2D6*9 allele, which has been shown to reduce catalytic activity, including bufuralol 1'-hydroxylation, compared to the WT CYP2D6*1 allele [15,28,30]. This aligns with our findings of low CYP2D6 activity observed in WT cells (Fig 1F and 1H).

To address this limitation, we used two distinct approaches: FLAG-tagged CYP2D6 cells and IRES-GFP co-expressing CYP2D6 HepaRG cells. As expected, introducing functional CYP2D6 genes into undifferentiated HepaRG cells effectively increased protein expression and enzyme activity (Fig 1E–1H). Furthermore, these cells enabled monitoring of CYP2D6 expression through FLAG and GFP signals, respectively (Fig 1D). These results suggest that the expression of FLAG and GFP in the transgenic cells can be used as an indicator of functional CYP2D6 expression.

To evaluate the differentiation capacity of genetically modified cells, we performed differentiation experiments to determine whether CYP2D6-enhanced HepaRG cells maintain their ability to differentiate into hepatocyte-like cells. Among the

five cell lines established in this study, only two retained the capacity to differentiate into CYP3A4-positive hepatocyte-like cells (Fig 3 and S5 Fig).

Modifying the genome of HepaRG cells presents several challenges, including the potential loss of differentiation capacity and the lack of hepatocyte-like characteristics. Previous studies indicate that overexpression of the constitutive androstane receptor, a key xenobiotic sensor in hepatocytes, promotes hepatic differentiation in human embryonic stem cells and HepaRG cells [24,31].

Our results revealed distinct differentiation patterns among the CYP2D6-enhanced cell lines. Specifically, the #1 FLAG-CYP2D6 and #1 CYP2D6-iGFP lines successfully retained their ability to differentiate into PAS-positive hepatocyte-like cells, whereas the remaining lines exhibited significantly reduced differentiation efficiency (Fig 3B and S4 Fig). Notably, the #1 CYP2D6-iGFP line, which exhibited the highest CYP2D6 activity, also maintained its differentiation potential, suggesting that elevated CYP2D6 expression does not directly cause reduced differentiation efficiency. Therefore, further studies are needed to evaluate the effects of genetic modification and enhanced CYP2D6 expression through comprehensive gene expression analysis.

CYP2D6 enzyme polymorphism significantly influences both the efficacy and toxicity of drugs metabolized by CYP2D6 in the human liver. Perhexiline, an antianginal drug that can cause drug-induced liver injury, is mainly detoxified by CYP2D6, which converts it to the less-toxic cis-OH-perhexiline [32]. Patients with CYP2D6 PM phenotypes, who have reduced mono-hydroxylation of perhexiline, are at increased risk for hepatotoxicity and neurotoxicity induced by perhexiline [33,34]. To confirm CYP2D6 function, we exposed undifferentiated HepaRG cells to perhexiline and assessed cytotoxicity. We found that perhexiline-induced toxicity was significantly reduced in #1 FLAG-CYP2D6 and #1 CYP2D6-iGFP cells compared to WT HepaRG cells (Fig 3B and 3C; S3 Fig). The estimated $IC_{50}$ values for perhexiline tended to increase with higher CYP2D6 activity in these cells (Fig 3D and S3 Fig). Meanwhile, cell viability exhibited an upward trend in #3 FLAG-CYP2D6 cells relative to WT controls, though this increase did not reach statistical significance. These findings are consistent with those reported by Ren et al. (2022), who demonstrated reduced perhexiline cytotoxicity in CYP2D6-overexpressing HepG2 cells while observing no protective effects with other CYP enzymes (CYP2C19, CYP1A2, CYP3A4) [32].

## 5 Conclusion

In summary, this study established HepaRG cell lines with enhanced CYP2D6 expression using FLAG-CYP2D6 and CYP2D6-iGFP vectors. The engineered cells showed increased CYP2D6 activity compared to WT cells, with the highest-expressing line (#1 CYP2D6-iGFP) reaching levels comprable to those in human liver tissue. CYP2D6 overexpression provided protection against perhexiline-induced cytotoxicity but not against CYP3A4-metabolized drugs. The CYP2D6-enhanced cell lines that retained differentiation capacity into hepatocyte-like cells while maintaining transgene expression, making them valuable cellular models for *in vitro* studies of drug metabolism and toxicity, particularly for compounds metabolized by CYP2D6.

## Supporting information

**S1 Fig. Raw images of Western blotting analysis.** (A and B) Uncropped raw images of Western blotting analysis related to Figure 1D (A) and Figure 2D (B). (C and D) Quantification of CYP2D6 expression normalized by b-actin in FLAG-CYP2D6 cell lines (C) and CYP2D6-iGFP cell lines (D).
(TIF)

**S2 Fig. Bufuralol dose-dependent hydroxylation by CYP2D6.** HepaRG cells were incubated with varying concentrations of bufuralol for 24 h. The amount of 1'-OH bufuralol produced was measured using LC-MS/MS analysis. BF indicates the bufuralol-treated group. DMSO was used as a vehicle control without a substrate.
(TIFF)

**S3 Fig. A cytotoxicity assay of perhexiline in FLAG-CYP2D6 HepaRG cells.** (A) Representative images of WT and FLAG-CYP2D6 HepaRG cells (#1 and #3) after 24-hour exposure to perhexiline. Phase-contrast micro-photographs are shown. Scale bar, 200 μm. (B) Relative cell viability of perhexiline-treated cells assessed by an ATP-based cell viability assay. Each dot represents an individual replicate (n = 3–4). P value calculated from Dunnett's multiple comparison test. *P < 0.05. (C) Dose-response curves of relative cell viability in perhexiline-treated cells. Each dot represents an individual value. The estimated IC50 value is indicated by a red closed circle. n = 3.
(TIFF)

**S4 Fig. Cytotoxicity assay of Amiodarone, Troglitazone, Tramadol and Trazodone in WT and #1 CYP2D6-iGFP HepaRG cells.** Relative viability of cells treated with known DILI-related compounds. Error bars, SD. n = 3.
(TIFF)

**S5 Fig. Hepatocyte differentiation in FLAG-CYP2D6 and CYP2D6-iGFP HepaRG cell lines.** (A) Representative images of PAS-stained cells in WT, FLAG-CYP2D6 and CYP2D6-iGFP HepaRG cell lines. PAS staining was performed at 28-day differentiation. Scale bar, 100 μm. (B) Representative images of PAS-stained cells in WT, #1 FLAG-CYP2D6, and #1 CYP2D6-iGFP HepaRG cells. After hepatic differentiation, PAS staining was conducted. Scale bar, 100 μm.
(TIFF)

**S6 Fig. Expression of CYP enzymes (CYP2D6, CYP1A2, and CYP3A4) in differentiated cells.** The gene expression levels of (A) CYP2D6 and (B) CYP1A2 in differentiated cells. Each dot represents an individual replicate (n = 3). (C) Representative immunofluorescence images showing the CYP3A4 expression in the differentiated #1 CYP2D6-iGFP cells. The enlarged area is outlined by the dotted square. Arrowheads indicate double-positive cells for CYP3A4 and GFP in differentiated #1 CYP2D6-iGFP cells. Scale bar, 100 μm.
(TIF)

## Acknowledgments

We sincerely thank Dr. Guguen-Guillouzo C., Dr. Jamin A. and Dr. Chesne C. (Biopredic International, France) for their expert advice on HepaRG cells. We would like to thank Dr. Muramoto T. and Mr. Asano R. for various suggestions and technical advice on this study. We also thank Mrs. Uekusa M. and Dr. Yamaguchi's laboratory members for routine maintenance of the laboratory and technical assistance.

## Author contributions

**Conceptualization:** Shinpei Yamaguchi, Chizuka Obara, Masako Tada.

**Data curation:** Chizuka Obara.

**Formal analysis:** Chizuka Obara.

**Funding acquisition:** Shinpei Yamaguchi.

**Investigation:** Shinpei Yamaguchi, Chizuka Obara, Yohei Iizaka, Akari Mine.

**Methodology:** Chizuka Obara.

**Supervision:** Shinpei Yamaguchi, Yojiro Anzai.

**Writing – original draft:** Chizuka Obara, Yohei Iizaka.

**Writing – review & editing:** Shinpei Yamaguchi.

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
