## [Decision Letter · Decision Letter 0]

15 Oct 2025

Dear Dr. Yamaguchi,

Thank you for submitting your manuscript to PLOS ONE. After careful consideration, we feel that it has merit but does not fully meet PLOS ONE’s publication criteria as it currently stands. Therefore, we invite you to submit a revised version of the manuscript that addresses the points raised during the review process.

We look forward to receiving your revised manuscript.

Kind regards,

Pramodkumar Pyarelal Gupta, PhD

Academic Editor

PLOS ONE

**Journal Requirements:**

1. When submitting your revision, we need you to address these additional requirements. Please ensure that your manuscript meets PLOS ONE's style requirements, including those for file naming. The PLOS ONE style templates can be found at https://journals.plos.org/plosone/s/file?id=wjVg/PLOSOne_formatting_sample_main_body.pdf and https://journals.plos.org/plosone/s/file?id=ba62/PLOSOne_formatting_sample_title_authors_affiliations.pdf 2. Thank you for stating the following financial disclosure: Toho University Grant for Research Initiative Program.   Please state what role the funders took in the study.  If the funders had no role, please state: "The funders had no role in study design, data collection and analysis, decision to publish, or preparation of the manuscript." If this statement is not correct you must amend it as needed. Please include this amended Role of Funder statement in your cover letter; we will change the online submission form on your behalf. 3. We note that you have included the phrase “data not shown” in your manuscript. Unfortunately, this does not meet our data sharing requirements. PLOS does not permit references to inaccessible data. We require that authors provide all relevant data within the paper, Supporting Information files, or in an acceptable, public repository. Please add a citation to support this phrase or upload the data that corresponds with these findings to a stable repository (such as Figshare or Dryad) and provide and URLs, DOIs, or accession numbers that may be used to access these data. Or, if the data are not a core part of the research being presented in your study, we ask that you remove the phrase that refers to these data. 4. Please include your full ethics statement in the ‘Methods’ section of your manuscript file. In your statement, please include the full name of the IRB or ethics committee who approved or waived your study, as well as whether or not you obtained informed written or verbal consent. If consent was waived for your study, please include this information in your statement as well. 5. PLOS ONE now requires that authors provide the original uncropped and unadjusted images underlying all blot or gel results reported in a submission’s figures or Supporting Information files. This policy and the journal’s other requirements for blot/gel reporting and figure preparation are described in detail at https://journals.plos.org/plosone/s/figures#loc-blot-and-gel-reporting-requirements and https://journals.plos.org/plosone/s/figures#loc-preparing-figures-from-image-files. When you submit your revised manuscript, please ensure that your figures adhere fully to these guidelines and provide the original underlying images for all blot or gel data reported in your submission. See the following link for instructions on providing the original image data: https://journals.plos.org/plosone/s/figures#loc-original-images-for-blots-and-gels.   In your cover letter, please note whether your blot/gel image data are in Supporting Information or posted at a public data repository, provide the repository URL if relevant, and provide specific details as to which raw blot/gel images, if any, are not available. Email us at plosone@plos.org if you have any questions. 6. If the reviewer comments include a recommendation to cite specific previously published works, please review and evaluate these publications to determine whether they are relevant and should be cited. There is no requirement to cite these works unless the editor has indicated otherwise. 

**Additional Editor Comments:**

In cell culture process the authors states here with following statement:

After 14 days, the cells were cultured in 1% DMSO medium for 2 days, and then further cultured in medium supplemented with 1.7% DMSO supplementation for 12 days. Is this standard protocol or an optimized kindly add appropriate reference to it.

Consistent with previous reports, the catalytic activity of CYP2D6 was extremely low in WT HepaRG cells, despite detectable protein levels in Western blot analysis.

what does it means here in terms of previous reports, do authors discussing about their previous published work? then add appropriate references.

As the work is carried out in Japan, the authors should consider PMDA guidelines too, in comparison to FDA Modernization Act 2.0.

the authors should significantly consider the other regulatory bodies guidelines too.

More detail statistical analysis is needed.

Add more references from the past data to support your work.

As per reviewer-1, additional experiment is suggested to carry out and support the work.

As per reviewer-2, add appropriate abbreviations, figure legends, image quality enhancement

Reviewers' comments:

**Comments to the Author**

1. Is the manuscript technically sound, and do the data support the conclusions?

Reviewer #1: Yes

Reviewer #2: Yes

2. Has the statistical analysis been performed appropriately and rigorously?

Reviewer #1: Yes

Reviewer #2: No

3. Have the authors made all data underlying the findings in their manuscript fully available?

Reviewer #1: Yes

Reviewer #2: Yes

4. Is the manuscript presented in an intelligible fashion and written in standard English?

Reviewer #1: Yes

Reviewer #2: Yes

**Reviewer #1: ** 1. Is the manuscript technically sound, and do the data support the conclusion?

- Yes

- Strength

- Technically acceptable and previously done by the same lab for CYP3A4 expression dynamics. Mentioned a reliable supply for the HepaRG cells and the culturing and enhancement methods are also appropriate. RT PCR and the western blotting, immunofluorescence staining, periodic Acid Schiff staining, CYP2D6 enzymatic activity and cytotoxicity assay were mentioned to have the suppliers and said to be done according to the manufacturer’s instructions and the experiments are appropriate to full fill the objectives of the study.

2. Has the statistical analysis been performed appropriately and rigorously?

- Yes

Strength

- Clear presentation of mean ± SD for continuous data, software disclosure good practice for transparency and reproducibility

- One way ANOVA and Dunne’s multiple comparison test are appropriate for multiple variant vs controls comparison

- P < 0.05 clarifies the significance level

- Technical triplicates per assay

- HepaRG control if possible empty vector control could also be included

To be clarified or corrected

- Assumption of normality and equal variance needs to be done and mentioned if it was done

- Multiple end points error, here several assays are running and to identify high expressing variants overall you might to include composite scoring or PCA

3. Have the authors made all data underlying the findings in their manuscript fully available?

- Yes

- The average measurements were presented in the figures at the apex. The descriptive statistics part should be presented as additional data; the means with SD but it is also possible to include median and interquartile range if the data is skewed is better to be considered.

4. Is the manuscript presented in an intelligible fashion and written in standard English?

- Yes

Strength

- The manuscript is presented with clear and standard English.

To be corrected or clarified

- Line 176,177,178 line spacing to be corrected

5. Review comments to the Author

- Generally, it is interesting to review a technically rich and thoroughly made study like this. The Authors has put great effort to make a significant contribution to the scientific community and to the existing body of knowledge. Technically the study is rich and the presentation also so much interesting. I am happy to read their manuscript to review it. It is well done and keep it up guys.

**Reviewer #2: ** An excellent piece of work. However, I suggest some following recommendations;

1) Abbreviate the short-terms used in manuscript. Authors please reassess and correct, throughout the text, as many of word are not abbreviated.

2) Check out recent PlosOne publications and format your manuscript accordingly, In-text/ end bibliography, manuscript sections and sub-headings, also assign numbers to headings and their sub-headings.

3) Kindly, arrange figure ligands and resolution according to Journal's format.

4) Include, Supplementary Table 1 and 2, into Main data.

5) How did you calculated p value, elaborate briefly.

6) which statistical model is used to design this experimental framework.

**Do you want your identity to be public for this peer review?** For information about this choice, including consent withdrawal, please see our Privacy Policy

Reviewer #1: **Yes: ** Berhan Ababaw

Reviewer #2: **Yes: ** Dr. Syeda Zahra Abbas Shah. Ph.D in Biotechnology ( Human Cancer Genomics, Molecular Biology and Computational Biology).

---

## [Author Response · Author response to Decision Letter 1]

22 Oct 2025

All point-by-point responses and exact manuscript changes are provided in the accompanying “Response to Editor and Reviewers.”

--

A. Responses to Journal Requirements

Response: We reformatted the manuscript to comply with PLOS ONE templates (title page, main text structure, figure legends, references, file names).

2. Please state what role the funders took in the study.

Response: We added the following statement to the cover letter and Funding section:

3. “Data not shown” not permitted

Response: All instances of “data not shown” were removed.

4. Ethics statement in Methods

Response: During our internal compliance check, we recognized that the diastase control used for PAS staining had been generated with fresh human saliva from one of the authors as an amylase source. Under our institutional policy and PLOS ONE’s ethics requirements, the use of human-derived material requires an IRB approval or an official exemption determination. Because retrospective determinations are generally not feasible and to avoid any ambiguity, we removed this dataset entirely (previous Figure S5B). The diastase-treated panel served only as an auxiliary confirmation for PAS staining and does not affect any analyses or the conclusions regarding enhanced CYP2D6 metabolic capacity. We have deleted all references to “diastase” from the Methods/legends and updated the figures accordingly.

5. Uncropped raw images for blots/gels

Response: We provide original, uncropped, unadjusted images for all blots as S1 Fig and reference them in figure legends.

6. Optional citation additions suggested by reviewers

Response: We evaluated suggested literature domains (HepaRG protocols, CYP2D6 in HepaRG, regulatory guidance context) and added relevant citations where appropriate (Line 112-113, reference 12, 21, and 22).

B. Responses to Additional Editor Comments

1. DMSO schedule (1% for 2 days → 1.7% for 12 days): standard or optimized? Add reference.

Response: We added relevant citations (Line 112-113, reference 12, 21, and 22).

2. “Consistent with previous reports, CYP2D6 catalytic activity is low in WT HepaRG despite detectable protein by western blot”—add references.

Response: We added relevant citations (Line 315, reference 15, 26, and 27).

3. Consider PMDA guidelines in comparison to the FDA Modernization Act 2.0; consider other regulatory bodies.

Response: We appreciate this suggestion. In the Discussion, we added a brief paragraph positioning our model within the policies of the Pharmaceuticals and Medical Devices Agency (PMDA) and the European Medicines Agency (EMA) regarding non-animal, human cell–based approaches, with appropriate citations (Line 436-441, reference 27).

4. More detailed statistical analysis; add references; address R1/R2 points.

Response: See Responses to Reviewers below.

5. As per reviewer-1, additional experiment is suggested to carry out and support the work. As per reviewer-2, add appropriate abbreviations, figure legends, image quality enhancement.

Response: We have addressed these points; details are provided in the point-by-point responses below.

C. Responses to Reviewer #1

We thank the reviewer for the careful evaluation and have clarified the strengths of our study where appropriate. We overlaid all individual measurements to enhance transparency of the descriptive statistics and corrected the noted line-spacing issue. Specific changes are described below.

1. HepaRG control if possible empty vector control could also be included

Response: We appreciate the suggestion and understand its rationale. Because our objective was to develop a CYP2D6-enhanced HepaRG platform, the appropriate negative comparator is the parental (WT) HepaRG line; adding an empty-vector control would mainly assess backbone effects and is unlikely to strengthen our platform-focused conclusions, so we did not add this experiment.

2. The descriptive statistics part should be presented as additional data; the means with SD but it is also possible to include median and interquartile range if the data is skewed is better to be considered.

Response: Thank you for this helpful suggestion. We revised the figures to overlay all individual measurements on each graph, so readers can directly assess the data distribution and variability (Fig 1F, H, I, Fig 2C, S4, S6 Fig A, B). These updates improve transparency and do not alter our conclusions.

3. Assumption of normality and equal variance needs to be done and mentioned if it was done

Response: We revised method section (Line246-247).

4. Line 176,177,178 line spacing to be corrected.

Response: We apologize for the line-spacing issue at lines 176–178; it has been corrected in the revised manuscript.

D. Responses to Reviewer #2

We are grateful for your thoughtful and practical suggestions on presentation. In response, we standardized abbreviations, revised all figure legends and image quality to match PLOS ONE format, moved Supplementary Tables 1–2 into the main text, and clarified how p values were calculated and which statistical model underlies our framework. Specific changes are described below.

1) Abbreviate the short-terms used in manuscript.

Response: We revised the manuscript to define all abbreviations at first mention and to use them consistently thereafter.

2) Check out recent PlosOne publications and format your manuscript accordingly, In-text/ end bibliography, manuscript sections and sub-headings, also assign numbers to headings and their sub-headings.

Response: We reformatted per PLOS ONE conventions; we adopted a numbered heading scheme consistently and standardized references and in-text citations.

3) Kindly, arrange figure legends and resolution according to Journal's format.

Response: We rearranged figure legends according to PLOS ONE's format.

4) Include, Supplementary Table 1 and 2, into Main data.

Response: We promoted Supplementary Table 1/2 into the main text as Table 1/2 (with concise summaries).

5) How did you calculated p value, elaborate briefly.

Response: We clarified that we report Dunnett-adjusted p values, with significance determined at α = 0.05 (two-sided). We revised method section (Line249-250).

6) which statistical model is used to design this experimental framework.

Response: We clarified that the framework is a fixed-effects one-factor linear model (group as the factor) with pre-specified Dunnett contrasts comparing each clone to the common WT control. Tests are two-sided with α = 0.05, and we report Dunnett-adjusted p values; the unit of analysis is biological replicates (technical replicates averaged). We revised method section (Line247-250).

---

## [Decision Letter · Decision Letter 1]

9 Dec 2025

Development of a CYP2D6-enhanced HepaRG Cell Model with Improved CYP2D6 Metabolic Capacity

PONE-D-25-31242R1

Dear Dr. Shinpei Yamaguchi

We’re pleased to inform you that your manuscript has been judged scientifically suitable for publication and will be formally accepted for publication once it meets all outstanding technical requirements.

Kind regards,

Pramodkumar Pyarelal Gupta, PhD

Academic Editor

PLOS One

Additional Editor Comments (optional):

As most of the comments made by the reviewers are positively addressed by the authors, the manuscript in further accepted.

Reviewers' comments:

Reviewer's Responses to Questions

**Comments to the Author**

Reviewer #2: All comments have been addressed

Reviewer #3: All comments have been addressed

2. Is the manuscript technically sound, and do the data support the conclusions?

Reviewer #2: Yes

Reviewer #3: Yes

3. Has the statistical analysis been performed appropriately and rigorously?

Reviewer #2: Yes

Reviewer #3: Yes

4. Have the authors made all data underlying the findings in their manuscript fully available?

Reviewer #2: Yes

Reviewer #3: Yes

5. Is the manuscript presented in an intelligible fashion and written in standard English?

Reviewer #2: Yes

Reviewer #3: Yes

Reviewer #2: I suggest editor to accept this manuscript. As all the queries have been addressed. So Congratulations to authors for their hard work.

Reviewer #3: The authors have addressed the comments and made the relevant changes to the manuscript. I would recommend that the authors organize the supplementary figures into a single PDF document with legends.

**Do you want your identity to be public for this peer review?** For information about this choice, including consent withdrawal, please see our Privacy Policy

Reviewer #2: **Yes: ** Syeda Zahra Abbas Shah

Reviewer #3: No

---

## [Editor Report · Acceptance letter]

PONE-D-25-31242R1

PLOS One

Dear Dr. Yamaguchi,

I'm pleased to inform you that your manuscript has been deemed suitable for publication in PLOS One. Congratulations! Your manuscript is now being handed over to our production team.

Kind regards,

on behalf of

Dr. Pramodkumar Pyarelal Gupta

Academic Editor

PLOS One